# Synchronous Head and Neck Cancer and Superficial Esophageal Squamous Cell Neoplasm: Endoscopic Treatment or No Treatment for the Superficial Esophageal Neoplasm

**DOI:** 10.3390/cancers15041079

**Published:** 2023-02-08

**Authors:** Chung-Wei Liu, Bo-Huan Chen, Chi-Ju Yeh, Cheng-Han Lee, Puo-Hsien Le, Yung-Kuan Tsou, Cheng-Tang Chiu

**Affiliations:** 1Department of Internal Medicine, Chang Gung Memorial Hospital, Taoyuan 33353, Taiwan; 2Department of Gastroenterology and Hepatology, Chang Gung Memorial Hospital, Taoyuan 33353, Taiwan; 3College of Medicine, Chang Gung University, Taoyuan 33353, Taiwan; 4Department of Pathology, Chang Gung Memorial Hospital, Taoyuan 33353, Taiwan

**Keywords:** head and neck cancer, superficial esophageal squamous cell neoplasm, synchronous, endoscopic resection, endoscopic submucosal dissection

## Abstract

**Simple Summary:**

How to treat patients with synchronous head and neck cancer and superficial esophageal squamous cell neoplasm (SHNSESCN) has not been studied. This study found that endoscopic resection (ER) of superficial esophageal squamous cell neoplasm (SESCN) significantly improved overall survival in patients with SHNSESCN compared with no treatment (NT) of SESCN. Furthermore, treatment-related mortality and morbidity were not significantly different between ER and NT of SESCN. Multivariate analysis showed that Eastern Cooperative Oncology Group performance status score and head and neck cancer disease progression were the two independent indicators of overall survival. Endoscopic resection of SESCN is the recommended treatment for patients with SHNSESCN, but further prospective randomized studies are needed to confirm this.

**Abstract:**

There are no studies on treating synchronous head and neck cancer (HNC) and superficial esophageal squamous cell neoplasm (SESCN). We aimed to report the outcomes of endoscopic resection (ER) and no treatment (NT) of SESCN in patients with synchronous HNC and SESCN (SHNSESCN). This retrospective study included 47 patients with SHNSESCN. Treatment for SESCN was ER (*n* = 30) or NT (*n* = 17). The ER group had significantly lower Charlson comorbidity index scores and a higher proportion of Eastern Cooperative Oncology Group performance status (ECOG PS) scores ≤1. The location and stage of the two tumors did not differ significantly between the groups. The 1-year, 3-year, and 5-year OS rates of the ER group were significantly better than those in the NT group. Treatment-related morbidity and mortality were not significantly different between the two groups. In the subgroup analysis of synchronous advanced HNC and SESCN, ER for SESCN also had a higher OS rate. Multivariate analysis showed that ECOG PS score and HNC disease progression were the two independent indicators of OS. In conclusion, treatment of SESCN with ER is the recommended approach for patients with SHNSESCN, but further randomized controlled trials are needed to confirm this.

## 1. Introduction

Head and neck cancer (HNC) is the sixth most common cancer worldwide; its incidence continues to rise, and it is anticipated to increase by 30% by 2030 [1]. Due to the field effect in cancer, it is not uncommon to observe a second primary esophageal squamous cell neoplasm (ESCN) in HNC patients [2,3]. Patients with synchronous HNC/ESCN have a worse prognosis than patients with isolated HNC [4,5]. Some studies have shown that ESCN is a prognostic indicator of overall survival (OS) in patients with synchronous HNC/ESCN [6,7]. Therefore, routine screening for ESCN is crucial for HNC patients [8,9]. With the advances in endoscopic diagnostic techniques and screening programs for HNC patients, superficial ESCN (SESCN, including Tis and T1 diseases) is increasingly detected [10,11]. There are treatment guidelines for isolated HNC or ESCN, but the management of synchronous HNC/ESCN remains unclear [12,13]. As a result, determining optimal treatment options for patients with these synchronous cancers is difficult. One-stage concurrent surgical resection and curative concurrent chemoradiotherapy (CCRT) have been reported to treat patients with non-metastatic synchronous HNC/ESCN [6,14]. However, it is unclear whether these two treatments are suitable for patients with synchronous HNC and SESCN (SHNSESCN), because esophagectomy and CCRT are too invasive for SESCN. Endoscopic resection (ER), including endoscopic mucosal resection (EMR) and endoscopic submucosal dissection (ESD), has recently become the treatment of choice for isolated SESCN because it is less invasive and is organ-sparing [13]. In addition, several recent reports suggest that ESD is non-inferior to esophagectomy in oncologic outcomes in isolated SESCN [15,16,17]. Whether ER in patients with SHNSESCN has a similar therapeutic effect remains to be elucidated. Therefore, this study aimed to report the outcomes of ER of SESCN versus no treatment (NT) of SESCN in patients with SHNSESCN. The primary outcome comparison was overall survival; secondary outcome comparisons were treatment-related mortality and morbidity.

## 2. Materials and Methods

### 2.1. Patient Selection

Between January 2008 and December 2017, 204 patients with cT1N0M0 SESCN identified from the computer database of our institution’s cancer registry were included in this study. Figure 1 shows the study flowchart. Synchronous neoplasms were defined as a second primary neoplasm diagnosed within 6 months of the first primary neoplasm diagnosis. The exclusion criteria were (1) patients without synchronous HNC (*n* = 136); (2) SESCN treated with CCRT (*n* = 10); (3) SESCN treated with esophagectomy (*n* = 5); (4) SESCN treated with radiofrequency ablation (*n* = 2); (5) lost to follow-up (*n* = 4). This study was reviewed and approved by the Ethics Committee of the Chang Gung Memorial Hospital (IRB No.: 202200880B0). Since this is a retrospective study using routine treatment or diagnostic medical records, the Chang Gung Medical Foundation Institutional Review Board approved the waiver of the participant’s consent.

### 2.2. Clinical Staging

For HNC, computed tomography (CT) scans and/or magnetic resonance imaging (MRI) of the head and neck were the primary tools used for staging. Integrated positron emission tomography and CT (PET-CT) scans were performed in stage III-IV patients. Stage I and II HNC were combined for early-stage disease, and stages III and IV were combined for advanced-stage disease.

For SESCN, endoscopic ultrasound (EUS) was performed to determine the depth of invasion: cT1a was a tumor involving the mucosa only; cT1b was a tumor involving the submucosa. In this study, both cT1a and cT1b ESCN were considered SESCN. Chest CT scans were the primary tool for N and M status assessment. PET-CT scans were performed for patients with pathologically proven cancer.

### 2.3. Treatments

In our institution, the management of HNC patients was decided by a dedicated multidisciplinary meeting involving otolaryngologists, oral surgeons, oncologists, radiation oncologists, radiologists, and pathologists. Our treatment guidelines for HNC are shown in Appendix A (guidelines were revised annually).

The management of patients with esophageal cancer was decided by another dedicated multidisciplinary meeting involving chest surgeons, gastrointestinal endoscopists, oncologists, radiation oncologists, radiologists, and pathologists. Treatment of patients with cTis SESCN did not necessarily have to be discussed at the meeting. For patients with SHNSESCN, the multidisciplinary esophageal team tended to recommend treating patients with HNC first, as previous clinical experience at our hospital had shown that some patients die from more advanced HNC. For patients with cTis SESCN patients, whether to treat SESCN is a matter of shared clinician–patient decision-making. The procedure of EMR was similar to the previous report and ESD to our previous publication [18,19]. Generally, in our clinical practice, only those patients with cT1a SESCN were recommended for ER. Only cT1b SESCN patients who refused or were not candidates for esophagectomy were considered for ER. None of the patients in the NT group received ESCN-specific therapy, including immunotherapy. However, 16 of 17 patients received chemotherapy for HNC (14 patients received primary CCRT and 2 patients received postoperative adjuvant chemotherapy). Among these 16 patients, 12 received the PUL chemotherapy regimen; the remaining 4 received the PF regimen (3 combined with Taxotere).

### 2.4. Follow-Up

Patient follow-up data were updated in August 2021, or until death. At this point, 28 patients had died. The remaining 19 patients were still under follow-up at our institution. For HNC, patients received ENT field check-ups every 1–3 months in the first year, every 2–4 months in the second year, every 3–6 months in the third year, and every 6–12 months thereafter. Patients who received adjuvant radiation therapy or CCRT received CT scans or MRIs 3 months after radiation therapy and then every 1 year for 5 years. If there was any clinically suspicious recurrence, CT scans, MRI, or PET-CT scans were performed. For SESCN treated with ER, endoscopy was performed 3 months after ER and every 6 months thereafter for at least 5 years. CT/MRI scans were on the same schedule as HNC.

### 2.5. Definitions in This Study

HNC disease progression was defined as any image (CT/MRI/PET-CT) showing disease progression during follow-up compared to the initial image at diagnosis, regardless of the initial treatment response. SESCN disease progression was defined as T stage ≥2 on EUS and N stage ≥1 or M stage = 1 on CT/MRI/PET-CT during follow-up.

OS was defined as the time from the diagnosis of HNC to death from any cause.

Criteria for treatment-related complications and their grading of severity were based on the Common Terminology Criteria for Adverse Events v5.0.

### 2.6. Statistical Analyses

The data of continuous variables were expressed as median and range and compared using the Mann–Whitney U test. The data of categorical variables were expressed as a number (%), and the chi-square test or Fisher’s exact test was used for comparison. Univariate and multivariate Cox proportional hazards models were used for OS analysis of patient and tumor variables. Only those variables with *p* < 0.1 in univariate analyses were entered into a multiple regression analysis for the OS. The Kaplan–Meier estimator was used to estimate the survival probability between groups, and the log-rank test was used to compare survival outcomes. A two-tailed *p*-value < 0.05 was considered statistically significant. Statistical analysis was performed using SPSS software (Version 22; SPSS, Inc., Chicago, IL, USA).

## 3. Results

A total of 47 patients who met the inclusion and exclusion criteria were included in the study. We divided patients who received ER for SESCN into the ER group (*n* = 30, including 3 EMR and 27 ESD); those who did not receive treatment for SESCN entered the NT group (*n* = 17). Patient and tumor characteristics are summarized in Table 1. The median age of the patients was 53 years (range, 35–71 years), and there was no significant difference between the two groups. All but one of the patients (NT group) were male. The median Charlson comorbidity index (CCI) score was 3 (range, 2–6) in the ER group and 3 (range, 2–8) in the NT group (*p* = 0.041). The proportion of the Eastern Cooperative Oncology Group (ECOG) performance status (PS) score ≤1 was significantly higher in the ER group (100% vs. 82.4%, *p* = 0.042). The most common location of HNC was the hypopharynx (44.7%), followed by the oral cavity (21.3%) and the oropharynx (19.1%). However, 10.6% of patients had multicentric HNC. The location of HNC did not differ significantly between the two groups. The most common site of SESCN was the middle thoracic esophagus (38.3%), followed by the lower esophagus (34.0%). However, 14.9% of patients had multifocal SESCN at diagnosis (defined as ≥2 lesions). The location of SESCN did not differ significantly between the two groups. Among the HNC clinical stages, 27.7% of the patients were stage I + II, 8.5% were stage III, and 63.8% were stage IV. There was no significant difference in the HNC clinical stage between the two groups (advanced stage, ER vs. NT: 70.0% vs. 76.5%, *p* = 0.752). Among SESCN clinical staging, 85.7% of patients had T1a (including Tis) and 14.3% had T1b. The difference in SESCN clinical staging was not significant between the two groups (*p* = 1). For HNC, 16 patients initially underwent surgery, 13 (43.3%) in the ER group and 3 (17.6%) in the NT group (*p* = 0.074). The remaining 31 patients initially received CCRT, 17 (56.7%) in the ER group and 14 (82.4%) in the NT group.

### 3.1. Treatment-Related Complications

Treatment-related complications are listed in Table 2. Grade 3–4 complications such as mucositis, dermatitis, neutropenia, thrombocytopenia, and anemia were not significantly different between the two groups. One patient in the ER group and one patient in the NT group experienced a treatment-related death (3.3% vs. 5.9%, *p* = 1). Seven patients were hospitalized for treatment-related complications, including three (10.0%) in the ER group and four (23.5%) in the NT group (*p* = 0.235). Upper aerodigestive tract stenosis occurred in eight patients, seven (23.3%) in the ER group and one (5.9%) in the NT group (*p* = 0.228).

### 3.2. The Outcomes of the Patients

Table 2 summarizes the patient outcomes. The median follow-up for survivors (*n* = 19) was 63.6 months. HNC disease progression occurred in 15 patients, 6 (20.0%) in the ER group and 9 (52.9%) in the NT group (*p* = 0.027). SESCN disease progression occurred in four patients, one (3.3%) in the ER group and three (17.6%) in the NT group (*p* = 0.128). The patient in the ER group with SESCN disease progression had pT1b disease and did not initially achieve R0 resection. Fourteen patients died of HNC, six (20.0%) in the ER group and eight (47.1%) in the NT group (*p* = 0.095). Two patients died of SESCN, one (3.3%) in the ER group and one (5.9%) in the NT group (*p* = 1). The 1-year OS (83.3% vs. 58.8%, *p* = 0.047), 3-year OS (66.7% vs. 35.3%, *p* = 0.019), and 5-year OS (43.3% vs. 11.8%, *p* = 0.044) were significantly better in the ER group than in the NT group.

The Kaplan–Meier survival curves for OS are presented in Figure 2. During the study period, OS was significantly better in the ER group (*p* = 0.044, Figure 2a). In the subgroup analysis of patients with advanced-stage HNC, OS was also significantly better in the ER group (*p* = 0.045, Figure 2b).

### 3.3. Factors Associated with Poor Prognosis

Table 3 shows the univariate and multivariate analyses of OS for the entire study population. Univariate analysis revealed that CCI score (per score increased, hazard ratio (HR) = 1.417; 95% CI: 1.067–1.883, *p* = 0.016), ECOG PS (score >1 vs. score ≤1, HR = 12.541; 95% CI: 3.034–51.095, *p* = 0.001), and HNC disease progression (yes vs. no, HR = 5.191; 95% CI: 2.302–11.705, *p* = 0.001) were significantly associated with unfavorable OS. Two factors reached marginal significance: clinical T-stage of SESCN (T1b vs. Tis+T1a, HR = 2.869; 95% CI: 0.966–8.515, *p* = 0.058) and treatment for SESCN (NT vs. ER, HR = 2.106; 95% CI: 0.980–4.522, *p* = 0.056). In the multivariate analysis, ECOG PS (score >1 vs. score ≤1, HR = 11.745; 95% CI: 1.786–77.251, *p* = 0.010) and HNC disease progression (HR = 4.492; 95% CI: 1.753–11.509) were the two independent prognostic factors.

## 4. Discussion

Previous studies have shown that ESCN but not HNC stage is a poor prognostic indicator of OS in patients with synchronous HNC/ESCN [6,7]. However, the prognosis of superficial and advanced ESCN is significantly different; to our knowledge, there are no reports in the literature of patients with SHNSESCN. Our study found that HNC but not SESCN disease progression was an independent poor prognostic indicator of OS in patients with SHNSESCN.

Therefore, the question of how to manage SESCN in patients with SHNSESCN remains open. Although one-stage concurrent surgical resection and reconstruction of synchronous HNC/ESCN may be promising for OS, it is highly invasive and complex, with a morbidity rate of 57.5%–94.1% and mortality rate of 0–5% [14,20,21]. Additionally, in one study, 52.9% of patients had two or more complications [14]. Synchronous CCRT for both tumors may be another option. Previous studies have shown that patients with locally advanced synchronous HNC/ESCN who receive synchronous CCRT have a worse prognosis than patients with isolated ESCC who receive CCRT [6,22]. This is because extensive radiation therapy for synchronous HNC/ESCN may have resulted in more adverse events that could lead to treatment interruptions [6,22]. Therefore, from the SESCN perspective, both concurrent surgery and synchronous CCRT appear to be too invasive for SHNSESCN patients.

ER, especially ESD, has recently been reported to be non-inferior to esophagectomy in view of oncological outcomes in patients with isolated SESCN [15,16,17]. Therefore, surgery or CCRT for HNC combined with ER for SESCN may be an effective and less invasive treatment for SHNSESCN. Watanabe et al. reported 183 hypopharyngeal cancer patients who underwent CCRT [23]. Thirty-three (18.0%) of their patients had synchronous SESCN receiving EMR (*n* = 12), CCRT (*n* = 15), and chemotherapy (*n* = 6). They found median survival was longer with EMR compared with CCRT and chemotherapy (46.4 months vs. 22.7 months vs. 8.0 months, respectively). Similarly, our study did reveal better OS in patients in the ER group. However, the main complication associated with ER (mainly ESD) was esophageal stricture, with an incidence of 23.3–33.3% in our and other studies [24]. Fortunately, this complication can be resolved by endoscopic dilations.

In our experiences, the presence of synchronous HNC did not interfere with the implementation of ER for ESCN. Except for esophageal stricture, there were no severe complications related to ER such as esophageal bleeding or perforation in our patients. Similarly, Moura et al. reported 47 patients with HNC (25.5% synchronous and 74.5% metachronous) and SESCN who underwent ESD for ESCN [24]. Compared with isolated ESCN, they found that HNC did not affect ESD-related procedure time, R0 resection rate, and adverse events.

This study has some limitations. First, this was a retrospective study from a single center, which has inherent shortcomings. However, due to the rare disease entity, the current study remains valuable, as this is the first to report SHNSESCN. Second, there might be selection bias between the two groups, as patients in the ER group had better ECOG PS and a lower incidence of HNC progression disease. Therefore, it should be interpreted with caution that patients who received ER had better OS than those who received NT. Third, in the current study, we cannot answer the question of which tumor should be treated first, or simultaneously. Matsumoto et al. reported that in patients with synchronous HNC/ESCN receiving staged treatment, delaying treatment of less advanced cancer may not adversely affect survival [25]. However, delaying treatment of SESCN may miss opportunities for ER.

## 5. Conclusions

In conclusion, based on the primary and secondary outcomes of this study, treatment of SESCN with ER is the recommended approach for patients with SHNSESCN. However, ECOG PS score and HNC disease progression are the two independent factors for prognosis. Further prospective randomized controlled trials are needed to confirm the benefit of ER for the treatment of SESCN in patients with SHNSESCN.

## Figures and Tables

**Figure 1 cancers-15-01079-f001:**
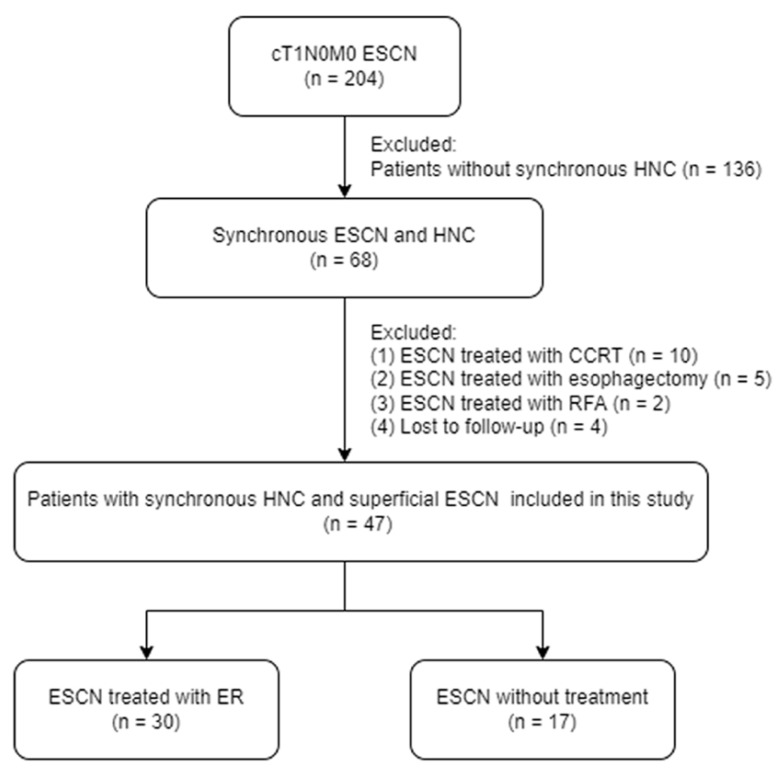
Flowchart of the study. Abbreviations: ESCN, esophageal squamous cell neoplasm; HNC, head and neck cancer; CCRT, concurrent chemoradiotherapy: RFA, radiofrequency ablation; ER, endoscopic resection.

**Figure 2 cancers-15-01079-f002:**
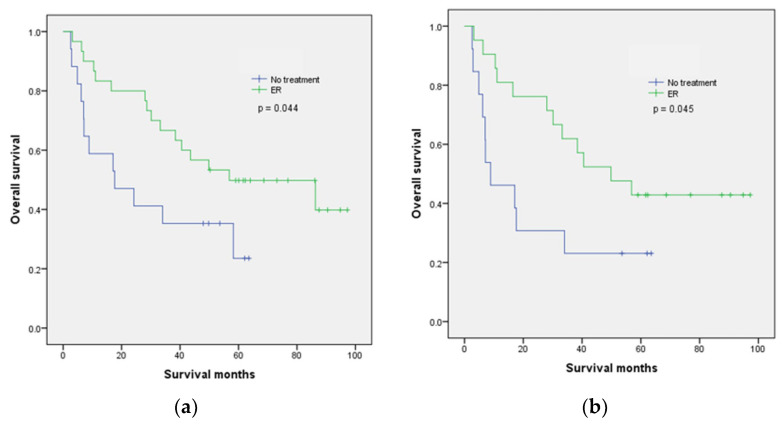
Kaplan–Meier survival curve shows the overall survival of the two groups: (**a**) overall patients; (**b**) advanced head and neck cancer patients.

**Table 1 cancers-15-01079-t001:** Patient and tumor characteristics.

	Overall (*n* = 47)	ER Group (*n* = 30)	No-Treatment Group (*n* = 17)	*p*-Value
Age, median (range), years	53 (35–71)	52 (45–68)	53 (35–71)	0.061
Sex, male	46 (97.9%)	30 (100%)	16 (94.1%)	0.362
CCI score, median (range)	3 (2–8)	3 (2–6)	3 (2–8)	0.041
ECOG PS score				0.042
≤1, *n*	44 (93.6%)	30 (100.0%)	14 (82.4%)	
>2, *n*	3 (6.4%)	0	3 (17.6)	
Location of HNC				0.412
Oropharynx, *n*	9 (19.1%)	5 (16.7%)	4 (23.5%)	
Hypopharynx, *n*	21 (44.7%)	14 (46.7%)	7 (41.2%)	
Larynx, *n*	1 (2.1%)	0	1 (5.9%)	
Oral cavity, *n*	10 (21.3%)	8 (26.7%)	2 (11.8%)	
Multicentric, *n*	5 (10.6%)	3 (10.0%)	2 (11.8%)	
Unknown of primary, *n*	1 (2.1%)	0	1 (5.9%)	
Location of SESCN				0.094
Cervical, *n*	1 (2.1%)	1 (3.3%)	0	
Upper third, *n*	5 (10.6%)	4 (13.3%)	1 (5.9%)	
Middle third, *n*	18 (38.3%)	13 (43.3%)	5 (29.4%)	
Lower third, *n*	16 (34.0%)	6 (20.2%)	10 (58.8%)	
Multiple, *n*	7 (14.9%)	6 (20.2%)	1 (5.9%)	
Clinical stage of HNC				0.752
Stage I + II, *n*	13 (27.7%)	9 (30.0%)	4 (23.5%)	
Stage III + IV, *n*	34 (72.3%)	21 (70.0%)	13 (76.5%)	
Stage III/IV, *n*	4/30	3/18	1/12	
Clinical T-stage of SESCN				1.000
Tis or T1a, *n*	43 (91.5%)	27 (90.0%)	16 (94.1%)	
T1b, *n*	4 (8.5%)	3 (10.0%)	1 (5.9%)	
Initial treatment for HNC				0.074
Surgery	16 (34.0%)	13 (43.3%)	3 (17.6%)	
CCRT	31 (66.0%)	17 (56.7%)	14 (82.4%)	

Abbreviation: ER, endoscopic resection; HNC, head and neck cancer; SESCN, esophageal squamous cell neoplasm; CCI, Charlson comorbidity index; ECOG PS, Eastern Cooperative Oncology Group Performance Status; CCRT, concurrent chemoradiotherapy.

**Table 2 cancers-15-01079-t002:** Treatment-related complications and outcomes.

	Overall (*n* = 47)	ER Group (*n* = 30)	No-Tx Group (*n* = 17)	*p*-Value
Treatment-related complications				
Hospitalized with complications, *n*	7 (14.9%)	3 (10.0%)	4 (23.5%)	0.235
Grade 5 ^†^, *n*	2 (4.3%)	1 (3.3%)	1 (5.9%)	1.000
Mucositis Grade 3–4 ^†^, *n*	6 (12.8%)	2 (6.7%)	4 (23.5%)	0.170
Dermatitis Grade 3–4 ^†^, *n*	8 (17.0%)	5 (16.7%)	3 (17.6%)	1.000
Neutropenia Grade 3–4 ^†^, *n*	5 (10.6%)	4 (13.3%)	1 (5.9%)	0.640
Anemia Grade 3–4 ^†^, *n*	8 (17.0%)	3 (10.0%)	5 (29.4%)	0.118
Thrombocytopenia Grade 3–4 ^†^, *n*	1 (2.1%)	0	1 (5.9%)	0.362
Upper aerodigestive tract stricture	8 (17.0%)	7 (23.3%)	1 (5.9%)	0.228
Outcomes				
Disease progression-HNC	15 (31.9%)	6 (20.0%)	9 (52.9%)	0.027
Disease progression-SESCN	4 (8.5%)	1 (3.3%)	3 (17.6%)	0.128
Died of HNC, *n*	14 (29.8%)	6 (20.0%)	8 (47.1%)	0.095
Died of SESCN, *n*	2 (4.3%)	1 (3.3%)	1 (5.9%)	1
All-cause mortality, *n*	28 (59.6%)	16 (53.3%)	12 (70.6%)	0.356
1-year overall survival	74.5%	83.3%	58.8%	0.047
3-year overall survival	55.3%	66.7%	35.3%	0.019
5-year overall survival	31.9%	43.3%	11.8%	0.044

Abbreviation: ER, endoscopic resection; HNC, head and neck cancer; SESCN, superficial esophageal squamous cell neoplasm. ^†^ According to National Cancer Institute Common Terminology Criteria for Adverse Events, Version 5.0.

**Table 3 cancers-15-01079-t003:** Univariate and multivariate analysis of overall survival for overall patients.

Variables	Univariate Analysis		Multivariate Analysis	
	HR (95% CI)	*p*-Value	HR (95% CI)	*p*-Value
Age, per year increase	0.996 (0.947–1.048)	0.892		
CCI score, per score increase	1.417 (1.067–1.883)	0.016	1.111 (0.809–1.525)	0.516
ECOG PS score				
≤1	1			
>1	12.541 (3.034–51.095)	0.001	11.745 (1.786–77.251)	0.010
Multicentric HNC at diagnosis				
Single	1			
Multiple	2.375 (0.813–6.940)	0.114		
Clinical stage of HNC				
Stage I + II	1			
Stage III + IV	1.890 (0.764–4.678)	0.169		
Clinical T-stage of SESCN				
Tis + T1a	1			
T1b	2.869 (0.966–8.515)	0.058	2.324 (0.708–7.631)	0.164
Treatment for HNC				
Surgery	1			
CCRT	1.707 (0.720–4.045)	0.224		
Treatment for SESCN				
ER	1			
No treatment	2.106 (0.980–4.522)	0.056	1.080 (0.438–2.667)	0.867
HNC disease progression				
No	1			
Yes	5.191 (2.302–11.705)	0.001	4.492 (1.753–11.509)	0.002
SESCN disease progression				
No	1			
Yes	1.340 (0.403–4.459)	0.663		

Abbreviations: HR, hazard ratio; CI, confidence interval; CCI, Charlson comorbidity index; ER, endoscopic resection; CCRT, concurrent chemoradiotherapy; C-T, chemotherapy; HNC, head and neck cancer; SESCN, superficial esophageal squamous cell neoplasm; CCI, Charlson comorbidity index; ECOG PS, Eastern Cooperative Oncology Group Performance Status.

## Data Availability

Deidentified individual participant data are available and will be provided on reasonable request to the corresponding author.

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
