# Peer review of "Synchronous Head and Neck Cancer and Superficial Esophageal Squamous Cell Neoplasm: Endoscopic Treatment or No Treatment for the Superficial Esophageal Neoplasm"

_cancers, 2023, doi:10.3390/cancers15041079_

Round 1

Reviewer 1 Report

Dear Editor:

Thank you for considering me as a reviewer for Cancers.

The Authors present an interesting paper on the treatment of synchronous head and neck cancer and superficial esophageal squamous cell neoplasms (SHNSESCN). It is an attractive topic in our area because these are anatomical locations that constitute a field of cancerization with common carcinogens, so these synchronous tumors are a challenge not very frequent, but common in our clinical area.

As mentioned by the Authors, the management of these cases is not clearly established, and they propose an interesting approach with an effective and less invasive surgical treatment than those existing in the past.

The paper is well structured, and the methodology used seems adequate. The comparison between patients treated with endoscopic resection and untreated patients may be interesting initially, although it could be of greater impact to evaluate the results in relation to treatment with concurrent chemoradiotherapy in tumors of the same stage.

In summary, I think that the present manuscript is suitable for publication in Cancers.

Author Response

Thank you for reviewing our manuscript and giving us some recommendations. We agree that CCRT can be one of the options for SESCN treatment in patients with SHNSESCN, especially when the lesion is located in the upper esophagus. However, due to the small number of cases in which patients with both tumors underwent CCRT, we were not able to analyze the outcomes of patients who underwent CCRT in this study.

Reviewer 2 Report

Review: “Synchronous head and neck cancer and superficial esophageal squamous cell neoplasm: endoscopic treatment or no treatment for the superficial esophageal neoplasm”

Dear authors,

Dear editor-in-chief, 

I read with interest the paper entitled “Synchronous head and neck cancer and superficial esophageal squamous cell neoplasm: endoscopic treatment or no treatment for the superficial esophageal neoplasm” and I would like to congratulate the authors for their effort. 

Here you can find a detailed stepwise review.

Aim of the paper:

-       To report the outcomes of endoscopic resection (ER) and no treatment (NT) of SESCN in patients with synchronous HNC and SESCN (SHNSESCN).

Strengths of the study:

-       There are no studies on treating synchronous head and neck cancer (HNC) and superficial esophageal squamous cell neoplasm (SESCN). 

-       There are treatment guidelines for isolated HNC or ESCN, but the management of synchronous HNC/ESCN remains unclear.

-       The exclusion criteria are precisely described.

-       The treatment guidelines, annually revised, for HNC are shown in a clear flowchart. 

-       “Disease progression”, “OS”, “treatment-related complications” are precisely defined.

-       Significant results. 

-       Clear tables about patient and tumor characteristics and treatment-related complications and outcomes. 

-       The statistical analyses are well described.

Major concerns: 

-       Retrospective study from a single center with small study cohort (47 patients).

-       Inclusion criteria, for example the time interval considered, are not specified. The minimum period of follow-up is unclear, too. Please specify.

-       It is not clear on what basis the multidisciplinary group decided whether to treat or not to treat the SESCN. Please specify.

-       Moreover, it is not clear if NT (no treatment group) was treated with either first-line chemotherapy and/or with immunotherapy. Please specify exactly the meaning. 

-       If the NT did not undergo any sort of treatment, why did the multidisciplinary group decided to do that?

-       “there might be selection bias between the two groups, as patients in the ER group had better ECOG PS and a lower incidence of HNC progression disease.” This is of course a limitation of the study and the authors properly mentioned it. I would suggest a future multicentric study (if possible) with paired matched analysis in order to provide more reliable data.

Article with related topic:

1.    Chung CS, Lo WC, Lee YC, Wu MS, Wang HP, Liao LJ. Image-enhanced endoscopy for detection of second primary neoplasm in patients with esophageal and head and neck cancer: A systematic review and meta-analysis. Head Neck. 2016 Apr;38 Suppl 1:E2343-9. doi: 10.1002/hed.24277. Epub 2015 Nov 23. PMID: 26595056.

2.    Kim DH, Gong EJ, Jung HY, Lim H, Ahn JY, Choi KS, Lee JH, Choi KD, Song HJ, Lee GH, Kim JH, Roh JL, Choi SH, Nam SY, Kim SY, Baek S. Clinical significance of intensive endoscopic screening for synchronous esophageal neoplasm in patients with head and neck squamous cell carcinoma. Scand J Gastroenterol. 2014 Dec;49(12):1486-92. doi: 10.3109/00365521.2013.832369. Epub 2014 Nov 5. PMID: 25372595.

3.    Gong EJ, Kim DH, Ahn JY, Choi KS, Jung KW, Lee JH, Choi KD, Song HJ, Lee GH, Jung HY, Kim JH, Roh JL, Choi SH, Nam SY, Kim SY. Routine endoscopic screening for synchronous esophageal neoplasm in patients with head and neck squamous cell carcinoma: a prospective study. Dis Esophagus. 2016 Oct;29(7):752-759. doi: 10.1111/dote.12404. Epub 2015 Oct 15. PMID: 26471351.

Self-references from the authors:

1.    Wang, C.Y.; Chen, B.H.; Lee, C.H.; Le, P.H.; Tsou, Y.K.; Lin, C.H. cT1N0M0 Esophageal Squamous Cell Carcinoma Invades the 321 Muscularis Mucosa or Submucosa: Comparison of the Results of Endoscopic Submucosal Dissection and Esophagectomy. Can- 322 cers (Basel) 2022, 14, doi:10.3390/cancers14020424. 

2.    Tsou, Y.K.; Chuang, W.Y.; Liu, C.Y.; Ohata, K.; Lin, C.H.; Lee, M.S.; Cheng, H.T.; Chiu, C.T. Learning curve for endoscopic 333 submucosal dissection of esophageal neoplasms. Dis Esophagus 2016, 29, 544-550, doi:10.1111/dote.12380.

Author Response

The authors would like to thank the reviewers for their valuable recommendations which greatly improved the manuscript. Regarding the main concerns of the reviewers, the reply is as follows:

  1. Retrospective study from a single center with small study cohort (47 patients).

A:  The authors agree that this is a shortcoming of the study. However, conducting prospective studies in only one center is difficult.

  1. Inclusion criteria, for example the time interval considered, are not specified. The minimum period of follow-up is unclear, too. Please specify.

A:  This retrospective study collected data from patients (with cT1N0M0 ESCN) from the computerized database of our institutional cancer registry between January 2008 and December 2017. Due to the rare disease entity, we included patients over a 10-year period. (revised on page 3 of the revised manuscript)

Patient follow-up data updated in August 2021. At this point, 28 patients had died. The remaining 19 patients were still under OPD follow-up at our institution. (revised on page 5 of the revised manuscript)

  1. It is not clear on what basis the multidisciplinary group decided whether to treat or not to treat the SESCN. Please specify.
  2. If the NT did not undergo any sort of treatment, why did the multidisciplinary group decided to do that?

A:  Since there is no research on whether to treat SHNSESCN patients with SESCN, according to our hospital's previous clinical experience, some patients will die of more advanced HNC. Therefore, multidisciplinary esophageal teams tend to recommend treating patients with HNC first. However, this recommendation is based on clinical experience rather than evidence, which is one of the reasons we conducted this study. It’s our policy that esophageal multidisciplinary conferences only discuss patients with esophageal cancer. There were 12 patients with cTis SESCN (or high-grade dysplasia) in the NT group. In these patients, whether to treat SESCN is a matter of shared clinician-patient decision-making. The reasons why the remaining 5 cT1a or cT1b ESCN patients did not receive treatment were: 1 patient died after HNC treatment; 1 patient had no endoscopic ESCN after HNC treatment; 1 patient refused to receive SESCN treatment; the other two patients were due to clinician-patient shared decision-making.

(revised on page 5 of the revised manuscript)

  1. Moreover, it is not clear if NT (no treatment group) was treated with either first-line chemotherapy and/or with immunotherapy. Please specify exactly the meaning.

A:  None of the patients in the NT group received ESCN-specific treatments, including immunotherapy. However, 16 of 17 patients received chemotherapy for HNC (14 patients received primary CCRT and 2 patients received postoperative adjuvant chemotherapy). Among these 16 patients, 12 patients received a PUL chemotherapy regimen which consisted of 4-6 cycles of cisplatin 50 mg/m2 on day 1, and oral tegafur 200mg or uracil and tegafur 300mg/m2/day on days 1 to 14, calcium folinate 15mg, four times a day on days 1 to 14, repeated every 2 weeks. The other 4 patients received PF regimen (3 patients with Taxotere): cisplatin 60-100 mg/m2 day 1, and 5-FU 600-1000 mg/m2/d for 3-5 days, repeated every 3-4 weeks.

(revised on page 5 of the revised manuscript)

  1. “there might be selection bias between the two groups, as patients in the ER group had better ECOG PS and a lower incidence of HNC progression disease.” This is of course a limitation of the study and the authors properly mentioned it. I would suggest a future multicentric study (if possible) with paired matched analysis in order to provide more reliable data.

A:  The authors agree with the reviewer’s recommendation.